# Galectin-3 and Pentraxin-3 as Potential Biomarkers in Chronic Coronary Syndrome and Atrial Fibrillation: Insights from a 131-Patient Cohort

**DOI:** 10.3390/ijms26104909

**Published:** 2025-05-20

**Authors:** Alexandru-Florinel Oancea, Paula Cristina Morariu, Maria Godun, Stefan Dorin Dobreanu, Alexandru Jigoranu, Miron Mihnea, Diana Iosep, Ana Maria Buburuz, Radu Stefan Miftode, Diana-Elena Floria, Raluca Mitea, Cristina Gena Dascalu, Daniela Maria Tanase, Irina-Iuliana Costache-Enache, Mariana Floria

**Affiliations:** 1Faculty of Medicine, University of Medicine and Pharmacy “Grigore T. Popa”, 700115 Iasi, Romania; alexandru.oancea@umfiasi.ro (A.-F.O.); morariu.paula-cristina@email.umfiasi.ro (P.C.M.); godun.maria-mihaela@d.umfiasi.ro (M.G.); stefan.dobreanu15@gmail.com (S.D.D.); jigoranu.alexandru@yahoo.ro (A.J.); miron_mihnea@d.umfiasi.ro (M.M.); diana.iosep@umfiasi.ro (D.I.); ana-maria.buburuz@umfiasi.ro (A.M.B.); radu-stefan.miftode@umfiasi.ro (R.S.M.); diana-elena.iov@d.umfiasi.ro (D.-E.F.); cdascalu_info@yahoo.com (C.G.D.); irina.costache@umfiasi.ro (I.-I.C.-E.); floria.mariana@umfiasi.ro (M.F.); 2Saint Spiridon Emergency Hospital, 700115 Iasi, Romania; 3Faculty of Medicine Victor Papilian, University of Lucian Blaga, 550169 Sibiu, Romania; daria.mitea@ulbsibiu.ro

**Keywords:** atrial fibrillation, chronic coronary syndrome, galectin-3, pentraxin-3, coronary ischemic disease, biomarkers

## Abstract

Chronic coronary syndrome (CCS) and atrial fibrillation (AF) are prevalent cardiovascular conditions between whom there is a dual relationship, with significant morbidity and mortality. Recent studies have highlighted the roles of galectin-3 and pentraxin-3 as potential biomarkers in coronary atherosclerosis, yet their specific interactions and implications in patients with CCS and AF remain underexplored. This proof-of-concept study aimed to evaluate the levels of galectin-3 and pentraxin-3 in a cohort of patients with CCS and AF. A total of 131 patients diagnosed with CCS or/and AF were stratified based on coronary stenosis severity (significant, S-CCS and nonsignificant, N-CCS coronary lesions) and arrhythmia burden. Blood samples were collected to measure serum levels of galectin-3 and pentraxin-3 using enzyme-linked immunosorbent assay (ELISA) techniques. Clinical data, including demographic information, comorbidities, medication use, and biological markers of systemic inflammation, were recorded. The galectin-3 value was more than double in patients with S-CCS compared with those with N-CCS (17.39 ± 4.459 ng/mL versus 7.49 ± 2.732 ng/mL, *p* < 0.001). Atrial fibrillation was not associated with statistically significant variations in galectin-3 values, neither overall nor separately in the group of S-CCS or N-CCS. However, pentraxin-3 values were similar in S-CCS compared with those with N-CCS (2839.18 ± 1521.639 pg/mL versus 2564.07 ± 1299.055 pg/mL, *p* = 0.417). These values were lower in patients with sinus rhythm, with a mean of 2469.91 ± 1253.782 pg/mL, and steadily increased in those with paroxysmal, persistent, and permanent AF, for whom they reached a mean of 3162.87 ± 1893.068 pg/mL. Elevated levels of galectin-3 appear to correlate with coronary stenosis severity and may inform future strategies for risk stratification, patients’ selection for invasive coronarography or therapeutic targeting in CCS.

## 1. Introduction

Chronic coronary syndrome (CCS) and atrial fibrillation (AF) are interrelated conditions that share key pathophysiological mechanisms—chronic inflammation, ischemia, and myocardial remodeling—contributing to high morbidity and mortality. Ischemia-driven structural and electrical changes can promote AF, while AF may worsen coronary artery disease through hemodynamic stress and increased oxygen demand. Given the complexity of CCS and AF coexistence, identifying reliable biomarkers is essential for early diagnosis, risk stratification, and tailored therapeutic interventions. Among the most promising candidates in this arena are galectin-3 and pentraxin-3, both of which have garnered significant attention for their roles in inflammation and atherosclerosis—two pivotal processes that underpin the pathophysiology of both CCS and AF [1,2,3].

Galectin-3, a member of the galectin family of glycan-binding proteins, is primarily expressed in activated macrophages, endothelial cells, and fibroblasts. Its role in inflammation is multifaceted, as it participates in various cellular processes, including cell adhesion, migration, and apoptosis. In the context of cardiovascular disease, galectin-3 is produced in response to myocardial injury and stress, often serving as a marker of adverse cardiac remodeling and heart failure. Elevated levels of galectin-3 have been linked to increased inflammation, fibrosis, and the progression of atherosclerosis. A study by Li et al. found that elevated galectin-3 levels were associated with the presence of CAD as well as coronary stability and complexity [4]. This study found that higher levels of galectin-3 were associated with worse clinical outcomes in patients with CAD, suggesting that galectin-3 could be a valuable biomarker for identifying patients at increased risk for adverse events as elevated galectin-3 levels may indicate increased myocardial stress and remodeling, which are critical factors in the progression of heart disease. By promoting the accumulation of immune cells in atherosclerotic plaques, galectin-3 contributes to the inflammatory milieu that underpins plaque instability, thereby heightening the risk of acute coronary events. Furthermore, its role in mediating fibroblast activation and collagen deposition underscores its significance in the development of myocardial fibrosis, a common consequence of chronic coronary ischemia. Regarding AF, Procyk et al. claimed that galectin-3 could be a useful biomarker in AF diagnosis, that its concentration is different between the types of AF (paroxysmal, persistent, and permanent), and finally, that it can predict the recurrence of AF in patients after ablation, but also postoperative atrial fibrillation (POAF) after cardiac surgery [5,6].

Pentraxin-3, another key player in the inflammatory cascade, is an acute-phase protein produced in response to pro-inflammatory stimuli such as cytokines and pathogen-associated molecular patterns. Unlike the liver-produced C-reactive protein (CRP), pentraxin-3 is secreted by various immune and non-immune cells, including endothelial cells and adipocytes, and plays a crucial role in the regulation of innate immunity and the resolution of inflammation. Clinical studies have so far provided contrasting results, highlighting the debated role of pentraxin-3 as an active mediator of endothelial dysfunction, atherosclerotic plaque vulnerability, and worse outcomes after ischemic events [7]. Therefore, substantial evidence suggests a dual role of pentraxin-3 as a modulator or amplifier of the innate immune response. A metanalysis noted that CAD patients with the highest pentraxin-3 level had an increased risk of all-cause mortality, cardiac death, and cardiac events, but elevated pentraxin-3 level appeared to not significantly increase the risk of cardiac events in the stable CAD subgroup [8]. On the other hand, pentraxin-3 may be involved in the stabilization of atherosclerotic plaques, potentially reducing the risk of rupture and subsequent thromboembolic events, as Ristagno et al. claimed that pentraxin-3 could have cardioprotective and atheroprotective roles regulating inflammation. Meanwhile, pentraxin-3 has also been implicated in AF; for instance, a study by Soeki et al. showed that local pentraxin-3 production in the left atrium might reflect the local inflammation of AF [9,10,11].

The balance between these two molecules may serve as a critical indicator of disease severity and progression. A predominance of galectin-3 over pentraxin-3 may suggest a shift toward a more inflammatory and destructive process, while a relative increase in pentraxin-3 could imply a protective response aimed at resolving inflammation. This nuanced interplay not only enhances our understanding of the pathophysiological mechanisms underlying CCS and AF but also opens avenues for exploring novel therapeutic strategies that target these biomarkers to modulate inflammation and improve clinical outcomes [5,12,13].

Systemic inflammation plays a key role in the progression of CCS and AF. Composite markers such as the Pan–Immune–Inflammation Value (PIV) and Systemic Immune–Inflammation Index (SII) reflect immune activation and thrombogenic potential. Elevated PIV—calculated as (neutrophils × platelets × monocytes)/lymphocytes—and SII—(neutrophils × platelets)/lymphocytes—have been associated with more severe coronary lesions and adverse cardiovascular outcomes, offering utility in risk stratification. Research focusing on NSTEMI patients found that a PIV cut-off of 568.2 predicted severe coronary lesions, with a sensitivity of 91% and specificity of 81.1% [14,15]. Another study involving CCS patients determined that an SII cut-off value of 620 was associated with 78.4% sensitivity and 64.0% specificity in predicting hemodynamically significant coronary artery stenosis [16]. Simpler ratios such as the lymphocyte-to-neutrophil (Ly/N) and lymphocyte-to-leukocyte (Ly/WBC) ratios also indicate systemic immune imbalance. Low Ly/N and Ly/WBC values suggest a pro-inflammatory state linked to atherosclerosis, AF development, and recurrence. A meta-analysis indicated that patients with a lower Ly/N had a notably higher rate of AF recurrence; in patients undergoing catheter ablation for AF, the rapport between neutrophils and lymphocytes predicted AF recurrence with a sensitivity of 73% and specificity of 67% [17,18].

This proof-of-concept study aimed to explore the association between galectin-3, pentraxin-3, systemic inflammatory markers, the severity of coronary stenosis, and AF burden in a cohort of patients with CCS and/or AF. We hypothesized that galectin-3 and pentraxin-3 levels, along with inflammatory indices, would differ according to coronary artery disease severity and arrhythmia presence, reflecting their potential utility in risk stratification.

## 2. Results

### 2.1. Baseline Characteristics

We enrolled 131 patients, further divided into two subgroups: 65 patients with S-CCS (significant chronic coronary syndrome defined as stenosis ≥ 70% for one of the coronary arteries, except for left main where stenosis ≥ 50%) at coronarography examination and 66 patients with N-CCS (nonsignificant chronic coronary syndrome defined as stenosis < 70% for coronary arteries, except for left main where stenosis < 50%). The N-CCS group also included the category of ANOCA (Angina with Non-Obstructive Coronary Arteries), which has garnered increasing attention because of its distinct pathophysiology and clinical implications. ANOCA patients often exhibit symptoms and ischemia despite the absence of obstructive coronary disease, suggesting underlying mechanisms such as microvascular dysfunction or vasospasm [19,20].

In Table 1, the general characteristics, comorbidities, treatment, echocardiographic, and laboratory parameters of the patients included are summarized. We found a significantly higher prevalence of important cardiovascular risk factors among patients S-CCS, such as dyslipidemia, diabetes mellitus, and a low level of HDL-cholesterol, compared with the baseline characteristics of the patients from the control group. However, regarding other traditional cardiovascular risk factors (age, gender, family history of ischemic coronary disease, smoking status, hypertension, obesity), there were no noteworthy differences between the two groups.

There was a significant difference (*p* = 0.026), with a higher percentage of sinus rhythm in the S-CCS group (53.8%) compared with the N-CCS group (37.9%). Moreover, AF was more prevalent in the N-CCS group (62.1%) compared with the S-CCS group (46.2%), with a *p*-value close to the statistically significant threshold (*p* = 0.067). The overall prevalence of PAD in this cohort was 9.9%, with a significant difference between the N-CCS group (4.5%) and the S-CCS group (15.4%), as indicated by a *p*-value of 0.038.

The overall mean of BMI (body mass index) is 30.35 kg/m^2^, with a statistically significant difference observed between the N-CCS group (31.27 kg/m^2^) and the S-CCS group (29.40 kg/m^2^), with a *p*-value of 0.031. However, despite the fact that the overall prevalence of obesity is high in both groups (77.1%), there is no significant difference between the two groups (*p* = 0.195). Smoking prevalence is similar across both groups (58.0% overall), with no significant difference (*p* = 0.648). There is a trend toward higher abdominal circumference in the N-CCS group (110.64 cm) compared with the S-CCS group (106.40 cm), but the difference is not statistically significant (*p* = 0.087).

An echocardiographic assessment of the LV systolic function showed a mean LVEF of 48.91%, with a standard deviation of 11.4%; there are no statistical differences between the two groups. Other mean values calculated based on echocardiographic data were as follows:Interventricular septum: 11.21 ± 1.663 mmLV posterior wall: 10.96 ± 1.459 mmLV end-diastolic diameter: 51.21 ± 7.081 mmLeft atrial indexed volume: 38.93 ± 14.292 mL/m^2^Right atrial indexed volume: 31.02 ± 11.715 mL/m^2^

A not statistically significant but suggestive difference (*p* = 0.050) in tricuspid regurgitation velocity suggests potential differences in right heart function between groups, but there are no differences between right ventricular longitudinal contractile function as there is no significant difference in tricuspid annular plane systolic excursion (*p* = 0.735).

Regarding biological parameters, there is a significant statistical difference (*p* = 0.004) for only high-density lipoprotein-cholesterol (HDLc), its levels being lower in the S-CCS group (40.88 mg/dL) compared with N-CCS (46.06 mg/dL), suggesting that this might be the most accurate parameter from lipid profile linked to coronary stenosis severity.

Marked differences in pharmacologic treatment were observed between the two groups. Among S-CCS patients, 60 out of 65 received dual antiplatelet therapy (aspirin and clopidogrel) following percutaneous coronary intervention (PCI), while the remaining five patients underwent coronary artery bypass grafting (CABG) and required SAPT. All patients who received DAPT were also prescribed PPIs in accordance with recommendations for gastroprotection during dual antiplatelet therapy. The consistent use of nitrates and trimetazidine in the S-CCS group further reflects comprehensive, guideline-directed management for patients with established CAD. The broader use of lipid-lowering agents in the S-CCS group underscores the emphasis on secondary prevention and the recognition of higher atherosclerotic risk in these patients.

### 2.2. Inflammatory Biomarkers and Parameters According to the Severity of Coronary Stenosis and the Presence/Type of AF

To assess the relationship between inflammation, AF, and coronary stenosis, patients were divided depending on CCS (N-CSS vs. S-CCS) and AF (sinus rhythm vs. the type of AF). Not only galectin-3 and pentraxin-3 were analyzed, but also CRP (C-reactive protein) and other inflammatory markers were included in this study, such as PIV, SII, Ly/N and Ly/WBC. The distribution of galectin-3 and pentraxin-3 levels according to the severity of coronary stenosis and the presence of AF is presented in Table 2.

The galectin-3 value is more than double in patients with S-CCS compared with those with N-CCS, with a mean of 17.39 ± 4459 versus 7.49 ± 2732, which is an important and statistically significant difference (*p* < 0.001). AF, however, is not associated with statistically significant variations in galectin-3 values, neither globally nor separately in the group of patients with S-CCS or N-CCS. Overall, patients with SR have a mean galectin-3 level of 11.40 ± 6482, which remains similar in those with persistent or permanent AF and increases by about three units in those with paroxysmal AF (to a mean of 14. 26 ± 6.111). In the group of N-CCS, galectin-3 values are decreased; thus, in those with SR, the mean value recorded is 6.66 ± 2.777 and increases by about two units in those with various types of AF. In the group of S-CCS, the mean value of galectin-3 in those with SR is significantly higher (16.78 ± 5.111) and also increases by about two units in patients with various types of AF. The values of galectin-3, depending on the severity of CCS and presence/type of AF, are detailed in Figure 1.

Pentraxin-3 values are also higher in S-CCS compared with N-CCS, with a mean of 2839.18 ± 1521.639 versus 2564.07 ± 1299.055. These values are also lower in patients with SR, with a mean of 2469.91 ± 1253.782, and steadily increase in those with paroxysmal, persistent, and permanent AF, for whom they reach a mean of 3162.87 ± 1893.068. In the group of N-CCS, the lowest values for pentraxin-3 are also found in those with SR, with a mean of 2256.81 ± 763.433, the highest values being observed in those with paroxysmal AF, with a mean of 3144.08 ± 1751.972. In the group of S-CCS, the mean value of pentraxin-3 in those with SR is 2712.06 ± 1631.495, decreases slightly in those with paroxysmal AF, to a mean of 2352.82 ± 659.705 and reaches a maximum for patients with permanent AF (3933.74 ± 1819.380); there are clear variations, but the statistical significance threshold is not reached. The values of pentraxin-3, depending on the severity of CCS and presence/type of AF, are detailed in Figure 2.

Table 3 summarizes the values of all other traditional inflammatory parameters included in this study, according to the severity of coronary stenosis and the presence of AF, such as PIV, SII, Ly/N, Ly/WBC, and CRP.

PIV values are also higher in S-CCS, with a mean of 44.76 ± 30.775, compared with those with N-CCS, with a mean of 37.40 ± 24.584, being not statistically significant but suggestive (*p* = 0.057). Also, the mean PIV values in patients with SR is 37.95 ± 29.637 and increases in those with various types of AF, reaching a maximum of 45.00 ± 28.786 in those with permanent AF, but differences that this time are not statistically significant. SII values are visibly higher in patients with S-CCS, with a mean of 681.14 ± 352.469, compared with those with N-CCS, with a mean of 621.13 ± 341.148. The highest values are also found in patients with paroxysmal AF, with a mean of 715.17 ± 359.110, and the lowest in those with SR, with a mean of 602.89 ± 355.585, while patients with persistent or permanent AF have mean values for SII intermediate between the extremes already highlighted. None of these differences, however, are statistically significant.

On the other hand, the Ly/N ratio is slightly higher in N-CCS, with a mean of 0.46 ± 0.205, compared with S-CCS (0.41 ± 0.185). Patients with SR have slightly higher values of this ratio than those with various types of AF, with a mean of 0.47 ± 0.194. In the group of N-CCS, the highest values for the Ly/N ratio are also observed in patients with SR, with a mean of 0.51 ± 0.190, and the lowest are found in patients with paroxysmal AF, with a mean of 0.32 ± 0.103. The variations thus found are in the neighborhood of the statistical significance threshold, being not statistically significant but suggestive (*p* = 0.060). The Ly/WBC ratio is also higher in patients with N-CCS, with a mean of 0.29 ± 0.088, compared with those with S-CCS (0.26 ± 0.079), being a trend toward statistical significance threshold (*p* = 0.063), and also in patients with SR, with a mean of 0.28 ± 0.083, compared with those with various types of AF, with a mean of about 0.26. The highest values of this ratio are found in patients with N-CCS and SR, with a mean of 0.31 ± 0.091, and the lowest in patients with S-CCS and permanent AF, with a mean of 0.22 ± 0.078.

Regarding CRP, mean values are slightly lower in patients with N-CCS (0.53 ± 0.772) compared with those with S-CCS (0.75 ± 1.105). These values are also lower in patients with SR, with a mean of 0.50 ± 0.717, or persistent AF, with a mean of 0.37 ± 0.211. The highest CRP values were found in patients with S-CCS and permanent AF (1.15 ± 1.898).

### 2.3. Diagnostic Performance of Galectin-3 in CCS

We introduced the parameters with statistically significant differences between the S-CCS and N-CCS (galectin-3, BMI, TRV, and HDLc) into a binary logistic regression model. The model was constructed using the Enter method. The Hosmer–Lemeshow split test indicates that the constructed model is viable, as its results are statistically insignificant (*p* = 1.000). Initially, the prediction accuracy for S-CCS was 55.2%, and at the end, it reached 97.7%, indicating that the identified predictors are useful. The model explains 95.8% of the variance in the diagnosis of S-CCS (Nagelkerke R^2^ = 0.958), thus confirming its usefulness, and has a sensitivity of 97.4% and a specificity of 97.9%. Of all predictors tested, only galectin-3 has a statistically significant role; the rest did not contribute significantly to the model. Increasing the level of galectin-3 by one unit increases the odds of significant lesion by 14.003-fold when the other predictors are held constant. These results suggest that galectin-3 may be a relevant biomarker for identifying patients with significant coronary stenosis (Table 4).

For galectin-3, the cut-off value identified is 11.63—the corresponding AUC coefficient is 0.998, which means an excellent discriminatory power, with a sensitivity of 97.4% and a specificity of 97.9%. For the other tested parameters, the power of discrimination is much lower, and the identified cut-off values have modest sensitivity and specificity coefficients (Table 5, Figure 3).

### 2.4. Galectin-3: Correlations with the Other Inflammatory Parameters

We analyzed the Pearson correlation for the parameters with statistically significant differences between the S-CCS and N-CCS (galectin-3, BMI, TRV, and HDLc) and also between galectin-3 and other inflammatory parameters. Galectin-3 is statistically significantly inversely correlated, albeit weakly, with HDLc, which supports the protective role of HDLc in atherosclerosis. There is also a moderate positive correlation between Galectin 3 and PIV, which is statistically significant (*p*-value = 0.009) with a Confidence Interval of [−0.459, −0.068]. Moreover, there is a weak negative correlation between galectin-3, Ly/N, and, respectively, Ly/WBC; although the relationship is not very strong, it is statistically significant, suggesting again the protective role of lymphocytes against inflammation and atherosclerosis (Table 6).

## 3. Discussion

This study aimed to explore the role of galectin-3 and pentraxin-3 as biomarkers in patients with CCS and AF, highlighting their potential for improving diagnostic accuracy in these overlapping conditions. Moreover, this study offers a comprehensive comparison between patients with N-CCS and S-CCS, highlighting notable differences in clinical characteristics, comorbidities, echocardiographic findings, biochemical profiles, and treatment approaches.

Interestingly, patients with S-CCS exhibited a significantly lower BMI compared with those with N-CCS. This observation aligns with the “obesity paradox”, wherein overweight or mildly obese individuals with established cardiovascular disease (CVD) may experience better prognoses than their normal-weight counterparts. Several studies have documented this phenomenon, suggesting that excess adiposity might confer protective benefits in certain CVD contexts. However, the underlying mechanisms remain a topic of ongoing research. This underscores the complexity of cardiovascular risk factors, where traditional metrics such as BMI may not fully capture the nuances of disease progression. Despite the lower BMI, S-CCS patients demonstrated a higher prevalence of dyslipidemia and PAD, both indicative of advanced systemic atherosclerosis [21,22].

In this study, SR was observed in 45.8% of patients, with a statistically significant difference between the S-CCS group (53.8%) and the N-CCS group (37.9%) (*p* = 0.026). This also suggests a higher prevalence of AF among patients with N-CCS, an observation that may initially appear counterintuitive. However, emerging evidence suggests that even non-obstructive coronary lesions can contribute to the development of AF. The LIFE-Heart Study demonstrated that patients with coronary artery sclerosis—defined as non-critical plaques with stenosis < 75%—had a higher prevalence of AF compared with those with obstructive CAD. Specifically, AF was present in 11.3% of patients with coronary artery sclerosis versus 4.7% in those with obstructive CAD. This counterintuitive finding may be explained by the role of atrial ischemia in AF pathogenesis. Non-obstructive plaques can still impair atrial perfusion, leading to structural and electrical remodeling that predisposes to AF. Additionally, patients with N-CCS may have a higher burden of comorbidities such as hypertension and diabetes, which are known risk factors for AF [23,24]. Moreover, the presence of AF in patients with less severe coronary lesions may be underrecognized. The CASS registry reported a low prevalence of AF (0.6%) among patients with angiographically proven CAD, suggesting potential underdiagnosis or underreporting. This underestimation underscores the need for vigilant screening for AF in patients with N-CCS, as early detection and management are crucial for preventing adverse outcomes [25].

Echocardiographic parameters were largely comparable between the N-CCS and S-CCS groups, with no statistically significant differences observed in LVDD, IVS, LVPW, or LVEF. Similarly, indexed left and right atrial volumes were not significantly different between groups, suggesting a comparable degree of atrial structural remodeling. Of note, TRV was marginally lower in the S-CCS group (2.12 ± 0.616 m/s) compared with the N-CCS group (2.33 ± 0.613 m/s), with a *p*-value of 0.050, nearing statistical significance. TRV is commonly used as a surrogate for estimating PAPs, and elevated values are associated with pulmonary hypertension and right-sided cardiac pressure overload. The slightly lower TRV in S-CCS patients may reflect more preserved right ventricular-pulmonary circulation dynamics or potentially a difference in left-sided filling pressures and pulmonary venous backflow. However, as PAPs and TAPSE did not differ significantly, the clinical relevance of this finding remains unclear. It is possible that patients with N-CCS may have subclinical diastolic dysfunction or occult pulmonary vascular disease contributing to this observation. Future studies with comprehensive hemodynamic assessments or advanced imaging techniques such as right heart catheterization or speckle-tracking echocardiography may help clarify this association.

The analysis of biochemical markers revealed that most parameters—including glycemic indices, renal function (creatinine and estimated glomerular filtration rate), hepatic enzymes (AST, ALT, and GGT), and other lipid fractions (LDLc, total cholesterol, and triglycerides)—did not significantly differ between the N-CCS and S-CCS groups. However, a notable exception was observed in HDLc levels, which were significantly lower in S-CCS patients (40.88 ± 10.848 mg/dL vs. 46.06 ± 11.395 mg/dL in N-CCS, *p* = 0.004). HDLc plays a critical anti-atherogenic role by promoting reverse cholesterol transport, exerting antioxidant and anti-inflammatory effects, and improving endothelial function. Reduced levels of HDLc have been consistently linked to increased cardiovascular risk and more extensive coronary atherosclerosis. In this study, the lower HDLc in the S-CCS group may reflect an impaired atheroprotective lipid profile contributing to the severity of coronary lesions. Additionally, HDLc is sensitive to systemic inflammation, oxidative stress, and metabolic dysfunction—all of which are frequently present in patients with significant CAD. These findings suggest that HDLc may serve as a particularly sensitive biomarker in differentiating patients at higher risk for significant coronary pathology, even when other lipid or metabolic markers appear within acceptable ranges. This emphasizes the importance of evaluating not only LDLc levels but also the qualitative aspects of the lipid profile when assessing cardiovascular risk and planning lipid-lowering therapy [26,27].

Our findings suggest a strong association between galectin-3 levels and the severity of coronary stenosis. Serum levels of galectin-3 were more than double that in patients with S-CCS compared with those with N-CCS, with a high statistical significance (*p* < 0.001). This aligns with previous findings, such as those by Li et al., who demonstrated galectin-3’s association with the presence, complexity, and severity of CAD, as well as its potential to predict adverse events because of its involvement in myocardial fibrosis and inflammation. In addition, galectin-3 has been identified as a marker of cardiac remodeling and progression to heart failure in various clinical settings, further reinforcing its diagnostic and therapeutic relevance in chronic ischemic conditions [4].

Interestingly, while galectin-3 correlated robustly with the severity of coronary lesions, it did not vary significantly between patients with and without AF. This finding diverges slightly from other studies, such as those by Procyk et al., which suggest galectin-3 levels may help distinguish AF subtypes and predict recurrence post-ablation or postoperatively. However, the absence of a significant trend in our cohort might be attributed to sample distribution or the underlying predominance of ischemic pathology over arrhythmogenic remodeling [6].

In contrast, pentraxin-3, an acute-phase protein with regulatory roles in inflammation and endothelial function, exhibited a progressive increase across AF subtypes, with the highest levels in patients with permanent AF. These trends, while not statistically significant, mirror the findings of Soeki et al., who described localized pentraxin-3 elevation in the left atrium during AF as a reflection of local inflammatory processes. In our study, pentraxin-3 values were lowest in patients in sinus rhythm—particularly in the N-CCS group—and increased progressively in those with paroxysmal, persistent, and permanent AF, reaching the highest mean levels in permanent AF. This pattern suggests that pentraxin-3 may reflect cumulative inflammatory burden and structural remodeling associated with AF chronicity. Interestingly, in the S-CCS group, a biphasic trend was observed, with a slight decrease in paroxysmal AF followed by a marked increase in permanent AF, possibly indicating variable inflammatory responses influenced by both rhythm status and coronary pathology. Although pentraxin-3 levels were generally higher in the S-CCS group, the variation was modest and nonsignificant, aligning with meta-analyses that report stronger associations of pentraxin-3 with acute coronary syndromes rather than stable CAD. These findings support the hypothesis that pentraxin-3 could serve as a dynamic marker of AF progression and inflammatory activity, particularly in patients with underlying structural heart disease [8,11].

The interplay between galectin-3 and pentraxin-3 observed in our study underscores their distinct yet potentially complementary roles in cardiovascular pathophysiology. Galectin-3 appears to predominantly reflect pro-inflammatory and profibrotic activity, contributing to coronary plaque progression, atrial fibrosis, and myocardial remodeling. Its elevation is closely tied to macrophage activation and tissue scarring, supporting its role as a marker of chronic inflammation and structural deterioration. In contrast, pentraxin-3 may signify a more nuanced, context-dependent immune response, potentially serving as a regulatory factor involved in vascular homeostasis, modulation of acute inflammation, and plaque stabilization. This dual behavior is consistent with findings by Melvin et al., who reported that while elevated levels of both galectin-3 and pentraxin-3 in patients with stable CAD were associated with increased cardiovascular risk, the biomarkers act through non-redundant pathways—galectin-3 reflecting fibrotic stress and pentraxin-3 suggesting an adaptive or compensatory immune mechanism. Our data support this distinction: galectin-3 was more prominently associated with the severity of coronary stenosis, whereas pentraxin-3 showed variable expression across AF subtypes, potentially reflecting dynamic endothelial or inflammatory activation rather than a fixed fibrotic state. Together, these markers may offer complementary insights into different aspects of the inflammatory and remodeling spectrum in CCS and AF. However, given the exploratory nature of this study and the lack of statistically significant trends for pentraxin-3, further validation is necessary before definitive clinical implications can be drawn [12,28].

Composite inflammatory indices, including the PIV, SII, Ly/N, and Ly/WBC, provide a comprehensive reflection of systemic immune activation and inflammatory burden. In the present study, elevated levels of PIV and SII were noted in S-CCS and AF subgroups, with values approaching statistical significance. These observations are consistent with the expanding body of literature supporting the prognostic relevance of such composite inflammatory markers in cardiovascular risk stratification. Prior investigations in both NSTEMI and stable CAD cohorts have demonstrated that heightened systemic inflammatory activity, as captured by these indices, correlates with adverse cardiovascular outcomes. Moreover, the observed inverse correlations between galectin-3 and HDLc, Ly/N, and Ly/WBC ratios further underscore the complex interplay between lipid metabolism, immune modulation, and vascular inflammation. These findings reinforce the clinical relevance of integrating systemic inflammatory biomarkers and lipid parameters into a multidimensional approach for cardiovascular risk assessment and underscore the pathophysiological interdependence of metabolic and immune pathways in the progression of atherosclerotic disease [14,16,29,30]. These observations are promising, but given the limited cohort size, they should be interpreted as hypothesis-generating. Future investigations should explore whether integrating these systemic markers with specific biomarkers such as galectin-3 and pentraxin-3 can improve clinical decision-making in larger, more diverse populations.

Finally, the diagnostic performance of galectin-3 was remarkable. A cut-off value of 11.63 ng/mL yielded both sensitivity and specificity above 97%, as confirmed by ROC analysis (AUC = 0.998). These results underscore its potential as a robust, non-invasive tool for identifying patients with significant coronary lesions and might aid in optimizing the use of coronary angiography. However, it is important to emphasize that these findings are derived from a single-center study with a modest sample size and, therefore, should be considered preliminary. While the high diagnostic accuracy observed—particularly the AUC of 0.998—indicates a strong discriminative signal in our dataset, such performance may reflect characteristics unique to this cohort and should be interpreted with caution.

### Study Limitations

This study has several limitations that must be acknowledged when interpreting the results. First, the research was conducted in a single-center setting, which inherently limits the generalizability of the findings. Patient populations, clinical practices, and resource availability may vary significantly across institutions and geographic regions, potentially affecting the external validity of the conclusions drawn. Second, the sample size, although adequate for initial exploratory analyses, remains relatively modest. With a total of 131 patients divided into two nearly equal groups (N-CCS and S-CCS), the statistical power to detect subtle but clinically meaningful differences may have been limited, particularly for parameters that approached significance. A larger cohort would enhance the robustness of the statistical analyses and allow for multivariable modeling to control for potential confounding variables.

Another notable limitation pertains to the measurement of emerging inflammatory and fibrosis biomarkers, specifically galectin-3 and pentraxin-3. Due to resource constraints, only a single determination of each biomarker was performed per patient, exclusively at the time of admission. This restricted sampling approach precludes assessment of temporal dynamics, treatment response, or correlation with long-term outcomes. Given the evolving role of galectin-3 and pentraxin-3 in cardiovascular pathophysiology—especially their associations with myocardial fibrosis, vascular inflammation, and prognosis in CAD—the lack of serial measurements represents a missed opportunity for a more nuanced analysis. However, we consider this small study as a pilot one. Therefore, based on these results we can now develop a larger one.

## 4. Materials and Methods

### 4.1. Study Design, Patients and Investigation

We conducted a prospective proof-of-concept study that included 131 consecutively enrolled patients admitted with an indication of coronarography exploration between January and June 2024. Patients over the age of 18, with or without AF, who presented an indication of coronarography, presenting signs of stable CAD (due to high-risk features such as a strong family history of CAD, hypertension, dyslipidemia, diabetes, obesity, or smoking, and who presented angina or positive non-invasive tests such as exercise stress test, echocardiography stress, and myocardial scintigraphy or who were identified with coronary stenosis detected at coronary computed tomography angiography), and who were with the ability to comprehend and accept an informed consent form were included in this research.

Patients under the age of 18 years, patients without signed informed consent, patients with acute myocardial infarction, chronic kidney disease with creatinine clearance below 30 mL/min/1.73 m^2^, hemodynamically significant valve diseases (more than mild severity), heart failure (HF) with an NYHA class 3 and 4, patients with thyroid and psychiatric disorders were excluded from this study.

### 4.2. Clinical Investigation and Data Collection

For the patients who were enrolled, we gathered clinical, biological, electrocardiographic (ECG), and echocardiographic data. Sex, age, body mass index (BMI), and abdominal circumference were included in the demographic characteristics. The patients’ smoking or non-smoking status was also assessed. We noted patient-associated comorbidities such as a history of AF and its type, hypertension (HTN), HF, diabetes mellitus (DM), chronic kidney disease (CKD) with clearance of creatinine between 60 and 30 mL/min/1.73 m^2^, chronic pulmonary obstructive disease (COPD), aortic atherosclerosis plaques, dyslipidemia, peripheral arterial disease (PAD), transient ischemic attack (TIA) and stroke. The coronary angiography was performed using an Azurion 7 Philips machine, and the severity of coronary stenosis was assessed visually or using FFR/IFR (Fractional Flow Reserve/Instantaneous Wave Free Ratio) when the situation required. Chronic treatment with statins, beta-blockers, angiotensin-converting enzyme (ACE) inhibitors, antidiabetic medication, oral anticoagulants, antiplatelet agents, or proton pump inhibitors (PPIs) in all patients was noted.

For the patients’ biological profiles, we gathered information on their galectin-3, pentraxin-3, total cholesterol, low-density lipoprotein-cholesterol (LDLc), high-density lipoprotein-cholesterol (HDLc), triglyceride levels and conducted liver function tests, including alanine aspartate transferase (AST), alanine aminotransferase (ALT), gamma-glutamyl-transferase (GGT), renal function and fasting blood sugar levels. Moreover, regarding inflammation status, c-reactive protein (CRP) was dosed; PIV, SII, Ly/N, and Ly/WBC were calculated for each patient from hemoleucogram components.

### 4.3. Statistical Analysis

Statistical analysis was performed using the SPSS version 29.0 software package (SPSS Inc., Chicago, IL, USA). For numerical data, standard descriptive statistics parameters were calculated (mean, standard deviation, minimum, maximum values, and median). To compare the arithmetic means of quantitative variables in two samples, we used the student’s *t*-test under the hypothesis that the values of the analyzed parameter follow a normal distribution (a fact that was tested beforehand using a goodness-of-fit test, such as the Kolmogorov–Smirnov test). When comparing the arithmetic means across multiple samples, which also adhere to normality, we employed the ANOVA (analysis of variance) test after first testing for homogeneity of variances using Levene’s test. In cases where the variances were not homogeneous, we used the Welch correction for the ANOVA test, which is known to be more robust. If the normal distribution of values is not met, we used the non-parametric Mann–Whitney test to compare a parameter between two samples and the non-parametric Kruskal–Wallis test for comparing the same parameter across three or more samples; both tests are based on the analysis of ranks assigned to the values of the parameter being compared. In the case of comparisons between multiple samples, when we identified statistically significant differences, we also employed post-hoc tests to locate the differences between the samples taken two at a time. For the ANOVA test, we used the post-hoc Tukey test (specifically its Tukey–Kramer correction for unbalanced samples of different sizes); for the Welch ANOVA test, we employed the Games–Howell post-hoc test, and for the non-parametric Kruskal–Wallis alternative, we used the Dunn post-hoc procedure with Bonferroni adjustment.

The simplest method for testing differences between the values of a qualitative variable across different groups is the Chi-squared test, which we also employed in this study. To investigate the association between two quantitative variables, we calculated the Pearson correlation coefficient r, along with its significance level and the corresponding 95% confidence interval, which allowed for characterizing the direction and strength of this association. The significance level we used was the standard 0.05. Values of the significance coefficient *p* < 0.05 were characterized as statistically significant, while values of *p* < 0.01 were characterized as highly statistically significant. In order to determine the diagnostic performance of the included parameters, we compared the areas under the curve (AUC) that resulted from receiver operating characteristic (ROC) analysis.

### 4.4. Ethics

We conducted this proof-of-concept study according to the ethical principles mentioned in the Declaration of Helsinki (revised in 2013). At admission, all patients signed a standard written informed consent in order to participate in this study. The research protocol was validated by the local Ethics Committees of both the University of Medicine and Pharmacy “Gr. T. Popa” (no. 352/9 October 2023) and of the St. Spiridon Emergency Clinical Hospital (no. 75/11 September 2023).

## 5. Conclusions

This pilot study provides preliminary evidence that galectin-3 is a potentially valuable biomarker associated with the severity of CCS. Its high diagnostic accuracy in this specific cohort highlights its potential clinical decision-making, particularly in stratifying patients for invasive diagnostic procedures, but caution is warranted when generalizing these findings.

Pentraxin-3, while less specific for coronary artery disease, may provide complementary insights in the context of AF, reflecting underlying atrial inflammation and disease chronicity. When combined, these biomarkers may offer a nuanced understanding of the inflammatory burden in patients with CCS and AF.

Altogether, our results support the hypothesis that combining fibrosis-related biomarkers with inflammatory indices (PIV, SII, Ly/N, and Ly/WBC) could enhance cardiovascular risk stratification. However, given the exploratory design and the limited statistical power, these findings should be interpreted as a proof-of-concept.

Future multicenter studies with larger, prospectively followed cohorts and robust multivariable modeling are needed to validate the diagnostic and prognostic value of galectin-3 and pentraxin-3 in CCS and AF. Furthermore, serial measurements of these biomarkers could provide insights into their temporal dynamics, treatment response, and relationship with outcomes—areas that remain unexplored in the present study.

## Figures and Tables

**Figure 1 ijms-26-04909-f001:**
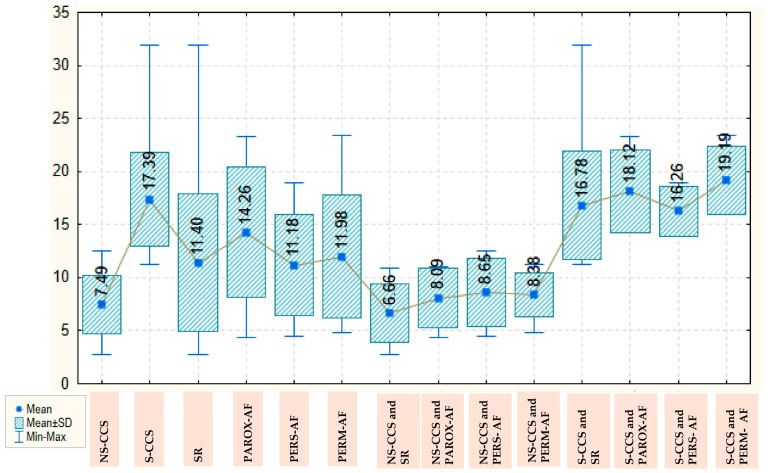
Box Plot of galectin-3 depending on the severity of chronic coronary syndrome and presence/type of atrial fibrillation. AF: atrial fibrillation; CCS: Chronic coronary syndrome; NS-CCS: nonsignificant chronic coronary syndrome; Parox-AF: paroxysmal AF; Pers-AF: persistent AF; Perm-AF: Permanent AF; S-CCS: significant chronic coronary syndrome; SR: sinus rhythm.

**Figure 2 ijms-26-04909-f002:**
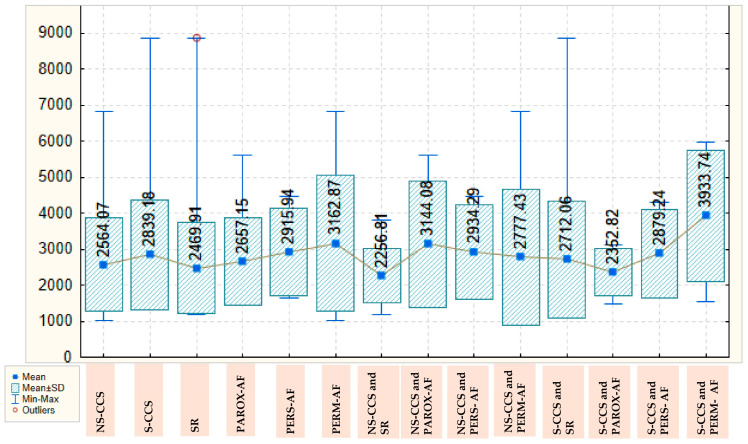
Box Plot of pentraxin-3 depending on the severity of chronic coronary syndrome and presence/type of atrial fibrillation. AF: atrial fibrillation; CCS: Chronic coronary syndrome; NS-CCS: nonsignificant chronic coronary syndrome; Parox-AF: paroxysmal AF; Pers-AF: persistent AF; Perm-AF: Permanent AF; S-CCS: significant chronic coronary syndrome; SR: sinus rhythm.

**Figure 3 ijms-26-04909-f003:**
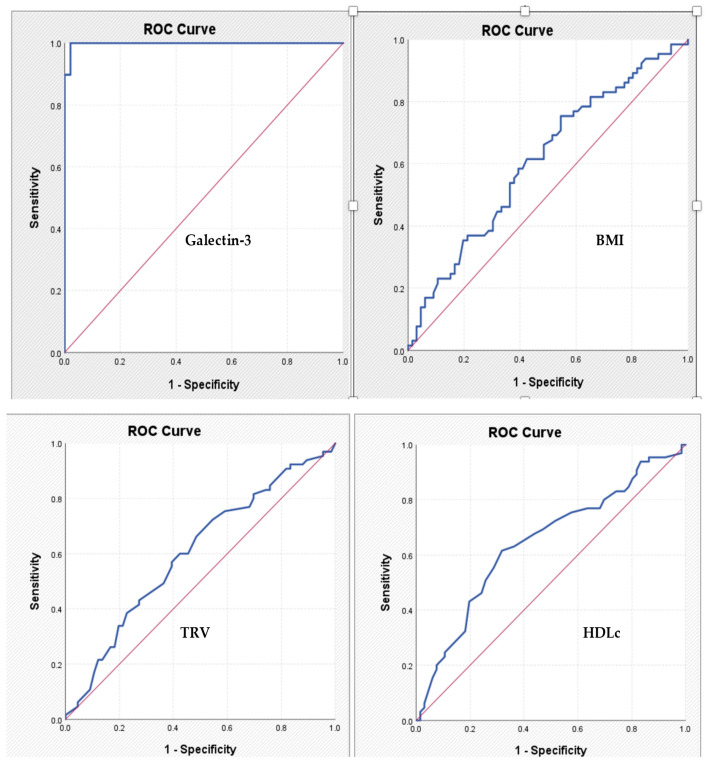
ROC curves expressing the association between specified parameters and the diagnosis of significant chronic coronary syndrome. BMI: body mass index; HDLc: high-density lipoprotein-cholesterol; TRV: tricuspid velocity.

**Table 1 ijms-26-04909-t001:** Demographic, clinical, laboratory, and echocardiographic characteristics of all patients included in this study (with and without significant chronic coronary lesions).

Parameter	Overall (131 pts)	N-CCS (66 pts)	S-CCS (65 pts)	*p*-Value
** *Clinical parameters* **				
Age (years)	66.46 ± 8.709	65.56 ± 9.806	67.37 ± 7.396	0.236
BMI (kg/m^2^)	30.35 ± 4.969	31.27 ± 4.965	29.40 ± 4.831	0.031 *
Male gender (%)	79 (60.3%)	38 (57.6%)	41 (63.1%)	0.520
Smoking (%)	76 (58.0%)	37 (56.1%)	39 (60.0%)	0.648
Abdominal circumference (cm)	108.53 ± 14.147	110.64 ± 15.067	106.40 ± 12.914	0.087
** *Comorbidities* **				
Family history of ischemic coronary disease (%)	77 (58.8%)	39 (59.1%)	38 (58.5%)	0.942
DM 2 (%)	47 (35.9%)	19 (28.8%)	28 (43.1%)	0.088
Arterial hypertension (%)	123 (93.9%)	60 (90.9%)	63 (96.9%)	0.274
Obesity (%)	101 (77.1%)	54 (81.8%)	47 (72.3%)	0.195
Dyslipidemia (%)	121 (92.4%)	57 (86.4%)	64 (98.5%)	0.017 *
PAD (%)	13 (9.9%)	3 (4.5%)	10 (15.4%)	0.038 *
CKD (%)	20 (15.3%)	9 (13.6%)	11 (16.9%)	0.601
COPD (%)	16 (12.2%)	6 (9.1%)	10 (15.4%)	0.217
Aortic atherosclerosis plaque (%)	93 (71.0%)	47 (71.2%)	46 (70.8%)	0.955
Epicardial fat (%)	55 (42.0%)	28 (42.4%)	27 (41.5%)	0.918
Sinus rhythm	60 (45.8%)	25 (37.9%)	35 (53.8)	0.026 *
AF (%)	71 (54.2%)	41 (62.1%)	30 (46.2%)	0.067
Prior stroke/TIA (%)	16 (12.2%)	10 (15.2%)	6 (9.2%)	0.301
** *Echocardiography parameters* **				
LVDD (mm)	51.21 ± 7.081	51.45 ± 7.506	50.97 ± 6.671	0.982
IVS (mm)	11.21 ± 1.663	11.32 ± 1.824	11.09 ± 1.487	0.562
LVPW (mm)	10.96 ± 1.459	10.94 ± 1.558	10.98 ± 1.364	0.776
LVEDV (mL)	136.54 ± 48.592	138.86 ± 55.791	134.18 ± 40.294	0.876
LVESV (mL)	72.92 ± 38.741	76.39 ± 47.535	69.39 ± 26.993	0.976
LVEF (%)	48.91% ± 11.4%	48.27 ± 12.484	49.57 ± 10.352	0.833
LA indexed volume (mL/m^2^)	38.93 ± 14.292	41.16 ± 16.422	36.67 ± 11.434	0.098
RA indexed volume (mL/m^2^)	31.02 ± 11.715	32.39 ± 13.493	29.63 ± 9.486	0.349
TRV (m/s)	2.22 ± 0.622	2.33 ± 0.613	2.12 ± 0.616	0.050 *
PAPs (mmHg)	26.51 ± 13.497	28.11 ± 14.216	24.89 ± 12.628	0.131
TAPSE (mm)	21.60 ± 4.007	21.48 ± 4.347	21.72 ± 3.659	0.735
** *Biological parameters* **				
Glycaemia (mg/dL)	118.34 ± 43.715	115.50 ± 36.457	121.23 ± 50.149	0.406
LDLc (mg/dL)	95.36 ± 37.081	97.95 ± 38.839	92.72 ± 35.311	0.485
HDLc (mg/dL)	43.49 ± 11.385	46.06 ± 11.395	40.88 ± 10.848	0.004 *
Total cholesterol (mg/dL)	158.16 ± 43.588	163.35 ± 43.563	152.89 ± 43.312	0.236
Triglycerides (mg/dL)	120.63 ± 60.578	115.67 ± 53.659	125.68 ± 66.922	0.443
AST (UI/L)	26.08 ± 15.414	27.64 ± 19.584	24.51 ± 9.384	0.750
ALT (UI/L)	25.98 ± 13.413	27.12 ± 16.310	24.82 ± 9.619	0.894
GGT (UI/L)	43.51 ± 43.789	48.92 ± 52.842	38.02 ± 31.593	0.099
Uric acid (mg/dL)	6.14 ± 1.577	6.37 ± 1.611	5.90 ± 1.518	0.089
Creatinine (mg/dL)	0.97 ± 0.284	0.93 ± 0.263	1.00 ± 0.301	0.101
GFR mL/min/1.73 m^2^	77.88 ± 21.050	80.62 ± 22.311	75.09 ± 19.465	0.075
** *Treatment* **				
ACEI/ARA-II (%)	118 (90.1%)	57 (86.4%)	61 (93.8%)	0.152
BB (%)	107 (81.7%)	54 (81.8%)	53 (81.5%)	0.967
Nitrates (%)	92 (70.2%)	27 (40.9%)	65 (100.0%)	<0.001 *
Trimetazidine (%)	77 (58.8%)	12 (18.2%)	65 (100.0%)	<0.001 *
Lipid-lowering drugs (%)	121 (92.4%)	57 (86.4%)	64 (98.5%)	0.017 *
PPIs (%)	69 (52.6%)	9 (13.6%)	60 (92.3%)	<0.001 *
DAPT (%)	60 (45.8%)	0 (0%0	60 (92.3%)	<0.001 *
SAPT (%)	28 (21.4%)	28 (42.4%)	0 (0%)	<0.001 *
Anticoagulation (%)	71 (54.2%)	41 (62.1%)	30 (46.2%)	0.067
Antiarrhythmics (%)	20 (15.3%)	10 (15.2%)	10 (15.4%)	0.970

* Statistical significance (*p* < 0.05); ACEI: Angiotensin-Converting Enzyme Inhibitors; AF: atrial fibrillation; ALT: Alanine aminotransferase; ARA-II: Angiotensin II receptor antagonists; AST: aspartate transferase; BB: betablockers; BMI: body mass index; CKD: Chronic kidney disease; COPD: Chronic obstructive pulmonary disease; DAPT: dual antiplatelet therapy; DM: diabetes mellitus; GGT: Gamma-glutamyl Transferase; GFR: glomerular filtration rate; HDLc: high-density lipoprotein-cholesterol; IVS: interventricular septum; LA: left atrium; LDLc: Low-density lipoprotein-cholesterol; LVDD: left ventricle diastolic diameter; LVEF: left ventricle ejection fraction; LVEDV: left ventricle end-diastolic volume; LVPW: left ventricle posterior (inferolateral) wall; LVESV: left ventricle end-systolic volume; N-CCS: nonsignificant chronic coronary syndrome; PAD: Peripheral artery disease; PAPs: estimated systolic pulmonary artery pression; PPIs: Proton pump inhibitors; RA: right atrium; SAPT: single antiplatelet therapy; S-CCS: significant chronic coronary syndrome; TAPSE: Tricuspid annular plane systolic excursion; TIA: transient ischemic attack; TRV: tricuspid regurgitation velocity.

**Table 2 ijms-26-04909-t002:** Galectin-3 and Pentraxin-3 values of all patients included in the study according to the severity of coronary stenosis and the presence of AF.

Parameter		N	Mean ± STD	Median (IQR:25–75)	*p*-Value
**Galectin-3 (ng/mL)**					
CCS	N-CCS	66	7.49 ± 2.732	7.50 (2.80–12.52)	<0.001 *
	S-CCS	65	17.39 ± 4.459	16.27 (11.30–31.90)	
AF	SR	60	11.40 ± 6.482	10.86 (2.80–31.90)	0.504
	Parox-AF	22	14.26 ± 6.111	14.37 (4.41–23.30)	
	Pers-AF	14	11.18 ± 4.724	10.69 (4.47–18.89)	
	Perm-AF	35	11.98 ± 5.774	9.79 (4.80–23.42)	
AF and N-CCS	SR	25	6.66 ± 2.777	6.55 (2.80–10.96)	0.181
	Parox-AF	8	8.09 ± 2.769	9.30 (4.41–10.98)	
	Pers-AF	9	8.65 ± 3.217	9.56 (4.47–12.52)	
	Perm-AF	24	8.38 ± 2.064	9.33 (4.80–11.29)	
AF and S-CCS	SR	35	16.78 ± 5.111	15.22 (11.30–31.90)	0.634
	Parox-AF	14	18.12 ± 3.929	17.92 (14.22–23.30)	
	Pers-AF	5	16.26 ± 2.327	15.42 (14.47–18.89)	
	Perm-AF	11	19.19 ± 3.239	17.56 (16.07–23.42)	
**Pentraxin-3 (pg/mL)**					
CCS	N-CCS	66	2564.07 ± 1299.055	2054.55 (1033.57–6833.04)	0.417
	S-CCS	65	2839.18 ± 1521.639	2363.64 (1466.46–8838.18)	
AF	SR	60	2469.91 ± 1253.782	2111.74 (1174.06–8838.18)	0.638
	Parox-AF	22	2657.15 ± 1198.978	2628.06 (1466.46–5611.47)	
	Pers-AF	14	2915.94 ± 1207.740	2363.64 (1645.26–4445.88)	
	Perm-AF	35	3162.87 ± 1893.068	2840.01 (1033.57–6833.04)	
AF and N-CCS	SR	25	2256.81 ± 763.433	2054.63 (1174.06–3805.10)	0.631
	Parox-AF	8	3144.08 ± 1751.972	1991.74 (1760.30–5611.47)	
	Pers-AF	9	2934.29 ± 1312.549	2703.52 (1645.26–4445.88)	
	Perm-AF	24	2777.43 ± 1883.519	2088.84 (1033.57–6833.04)	
AF and S-CCS	SR	35	2712.06 ± 1631.495	2150.16 (1565.54–8838.18)	0.414
	Parox-AF	14	2352.82 ± 659.705	2662.06 (1466.46–3116.70)	
	Pers-AF	5	2879.24 ± 1234.727	2363.64 (1985.88–4288.20)	
	Perm-AF	11	3933.74 ± 1819.380	4355.46 (1548.58–5962.95)	

* Statistical significance (*p* < 0.05); AF: atrial fibrillation; CCS: Chronic coronary syndrome; N-CCS: nonsignificant chronic coronary syndrome; Parox-AF: paroxysmal AF; Pers-AF: persistent AF; Perm-AF: Permanent AF; S-CCS: significant chronic coronary syndrome.

**Table 3 ijms-26-04909-t003:** Inflammatory parameters of all patients included in this study according to the severity of coronary stenosis and the presence of AF.

Parameter		N	Mean ± STD	Median (IQR:25–75)	*p*-Value	
**PIV = N × T × M/Ly × 10^7^**						
CCS	N-CCS	66	37.40 ± 24.584	29.18 (4.41–140.62)	0.057 ^+^	
	S-CCS	65	44.76 ± 30.775	36.67 (5.30–206.67)		
AF	SR	60	37.95 ± 29.637	29.49 (4.41–206.67)	0.280	
	Parox-AF	22	43.46 ± 22.133	35.43 (16.09–106.91)		
	Pers-AF	14	40.69 ± 27.927	32.42 (6.96–98.10)		
	Perm-AF	35	45.00 ± 28.786	41.78 (6.28–140.62)		
AF and N-CCS	SR	25	31.94 ± 20.030	26.21 (4.41–89.13)	0.487	
	Parox-AF	8	40.88 ±19.446	34.17 (25.04–85.72)		
	Pers-AF	9	35.86 ± 22.456	25.25 (7.29–74.11)		
	Perm-AF	24	42.51 ± 30.563	36.28 (6.28–140.62)		
AF and S-CCS	SR	35	42.25 ± 34.574	34.42 (5.30–206.67)	0.331	
	Parox-AF	14	44.93 ± 24.110	37.10 (16.09–106.91)		
	Pers-AF	5	49.39 ± 37.136	33.34 (6.96–98.10)		
	Perm-AF	11	50.45 ± 4.929	47.79 (30.09–119.72		
**SII = N × T/Ly × 10^3^**						
CCS	N-CCS	66	621.13 ± 341.148	533.23 (133.61–2130.57)	0.302	
	S-CCS	65	681.14 ± 352.469	596.25 (212.38–2348.52)		
AF	SR	60	602.89 ± 355.585	510.16 (133.61–2348.52)	0.194	
	Parox-AF	22	715.17 ± 359.110	681.64 (328.27–1749.46)		
	Pers-AF	14	681.16 ± 226.613	669.88 (314.71–1078.05)		
	Perm-AF	35	680.72 ± 364.807	636.80 (159.23–2130.57)		
AF and N-CCS	SR	25	524.97 ± 273.352	467.01 (133.61–1153.60)	0.115	
	Parox-AF	8	773.98 ± 421.917	700.16 (372.86–1749.46)		
	Pers-AF	9	653.87 ± 174.540	639.86 (370.35–897.50)		
	Perm-AF	24	658.08 ± 407.602	582.31 (159.23–2130.57)		
AF and S-CCS	SR	35	658.55 ± 398.765	524.44 (212.38–2348.52)	0.673	
	Paroxysmal AF	14	681.56 ± 330.337	633.20 (328.27–1527.25)		
	Pers-AF	5	730.30 ± 318.239	775.34 (314.71–1078.05)		
	Perm-AF	11	730.13 ± 257.778	770.81 (310.18–1209.25)		
**Ly/N**						
CCS	N-CCS	66	0.46 ± 0.205	0.441 (0.11–1.42)	0.144	
	S-CCS	65	0.41 ± 0.185	0.386 (0.13–1.18)		
AF	SR	60	0.47 ± 0.194	0.44 (0.18–1.18)	0.159	
	Parox-AF	22	0.39 ± 0.160	0.37 (0.13–0.89)		
	Pers-AF	14	0.39 ± 0.094	0.37 (0.24–0.57)		
	Perm-AF	35	0.42 ± 0.240	0.37 (0.11–1.42)		
AF and N-CCS	SR	25	0.51 ± 0.190	0.50 (0.24–0.97)	0.060^+^	
	Parox-AF	8	0.32 ± 0.103	0.34 (0.13–0.48)		
	Pers-AF	9	0.41 ± 0.088	0.40 (0.31–0.57)		
	Perm-AF	24	0.46 ± 0.256	0.40 (0.11–0.42)		
AF and S-CCS	SR	35	0.44 ± 0.195	0.41 (0.18–1.18)	0.164	
	Paroxysmal AF	14	0.42 ± 0.178	0.39 (0.13–0.89)		
	Pers-AF	5	0.35 ± 0.100	0.36 (0.24–0.50)		
	Perm-AF	11	0.34 ± 0.185	0.28 (0.15–0.72)		
**Ly/WBC**						
CCS	N-CCS	66	0.29 ± 0.088	0.28 (0.09–0.52)	0.063 ^+^	
	S-CCS	65	0.26 ± 0.079	0.25 (0.10–0.49)		
AF	SR	60	0.28 ± 0.083	0.28 (0.14–0.52)	0.541	
	Parox-AF	22	0.26 ± 0.088	0.25 (0.11–0.49)		
	Pers-AF	14	0.26 ± 0.056	0.25 (0.11–0.49)		
	Perm-AF	35	0.26 ± 0.094	0.25 (0.09–0.51)		
AF and N-CCS	SR	25	0.31 ± 0.091	0.31 (0.17–0.52)	0.376	
	Parox-AF	8	0.27 ± 0.089	0.24 (0.17–0.42)		
	Pers-AF	9	0.26 ± 0.036	0.25 (0.22–0.32)		
	Perm-AF	24	0.28 ± 0.097	0.26 (0.09–0.51)		
AF and S-CCS	SR	35	0.27 ± 0.074	0.26 (0.09–0.51)	0.487	
	Paroxysmal AF	14	0.26 ± 0.091	0.25 (0.11- 0.49)		
	Pers-AF	5	0.26 ± 0.087	0.24 (0.17–0.39)		
	Perm-AF	11	0.22 ± 0.078	0.20 (0.10–0.35)		
**CRP (mg/dL)**						
CCS	N-CCS	66	0.53 ± 0.772	0.30 (0.04–5.30)	0.490	
	S-CCS	65	0.75 ± 1.105	0.35 (0.03–6.55)		
AF	SR	60	0.50 ± 0.717	0.29 (0.03–4.37)	0.227	
	Parox-AF	22	0.88 ± 1.285	0.29 (0.04–1.37)		
	Pers-AF	14	0.37 ± 0.211	0.37 (0.11–0.85)		
	Perm-AF	35	0.83 ± 1.190	0.47 (0.03–6.55)		
AF and N-CCS	SR	25	0.36 ± 0.322	0.29 (0.04–1.37)	0.102	
	Parox-AF	8	0.80 ± 1.820	0.14 (0.10–5.30)		
	Pers-AF	9	0.37 ± 0.239	0.32 (0.12–0.85)		
	Perm-AF	24	0.69 ± 0.676	0.51 (0.08–2.41)		
AF and S-CCS	SR	35	0.60 ± 0.891	0.28 (0.03–4.37)	0.579	
	Paroxysmal AF	14	0.92 ± 0.938	0.38 (0.04–2.55)		
	Pers-AF	5	0.39 ± 0.173	0.45 (0.11–0.54)		
	Perm-AF	11	1.15 ± 1.898	0.46 (0.03–6.55)		

^+^ Very closed to statistical significance (*p* < 0.05); AF: atrial fibrillation; CCS: Chronic coronary syndrome; CRP: C reactive protein; Ly: Lymphocytes; M: monocytes; N: Neutrophils; N-CCS: nonsignificant chronic coronary syndrome; Parox-AF: paroxysmal AF; Pers-AF: persistent AF; Perm-AF: Permanent AF; PIV: Pan–Immune–Inflammation Value; S-CCS: significant chronic coronary syndrome; SII: Systemic immune–inflammation index; T: thrombocytes; WBC: leukocytes.

**Table 4 ijms-26-04909-t004:** Multivariate analysis: Galectin-3 predictor of significant chronic coronary syndrome.

Test Result Variable (s)	B	S.E.	Wald	df	Sig.	Exp(B) = OR	95% C.I. for EXP(B)	
							Lower	Upper
Galectin-3	2.639	1.015	6.760	1	0.009 *	14.003	1.915	102.398
Constant	−30.947	11.759	6.926	1	0.008	0.000		

* Statistical significance where *p* < 0.05.

**Table 5 ijms-26-04909-t005:** Detailed AUC, cut-off value, sensibility, and specificity for the specified parameters.

Test Result Variable (s)	Area Under the Curve				Cut-Off Value	Sensibility	Specificity
	Area	*p*-Value	Asymptotic 95% Confidence Interval				
			Lower Bound	Upper Bound			
BMI	0.610	0.029 *	0.514	0.707	31.62	0.754	0.455
Galectin-3	0.998	0.000 *	0.993	1.000	11.63	0.974	0.979
TRV	0.602	0.044 *	0.505	0.699	2.45	0.723	0.455
HDLc	0.645	0.049 *	0.550	0.740	41.50	0.615	0.682

* Statistical significance where *p* < 0.05; BMI: body mass index; HDLc: high-density lipoprotein-cholesterol; TRV: tricuspid velocity.

**Table 6 ijms-26-04909-t006:** Pearson Correlation between galectin-3 and other inflammatory parameters.

Confidence Intervals				
Parameters	Pearson Correlation r	*p*-Value	95% Confidence Interval	
			Lower	Upper
Galectin-3 and BMI	−0.140	0.196	−0.341	0.073
Galectin-3 and TRV	−0.096	0.374	−0.301	0.117
Galectin-3 and HDLc	−0.245	0.022 *	−0.433	−0.036
Galectin-3 and Pentraxin-3	0.050	0.643	−0.161	0.258
Galectin-3 and SII	0.065	0.547	−0.147	0.272
Galectin-3 and PIV	0.275	0.009 *	−0.459	−0.068
Galectina-3 and Ly/N	−0.215	0.044 *	−0.407	−0.005
Galectina-3 and Ly/WBC	−0.138	0.200	−0.339	0.074

* Statistical significance where *p* < 0.05; BMI: body mass index; HDLc: high-density lipoprotein-cholesterol; Ly: Lymphocytes; N: Neutrophils; PIV: Pan–Immune–Inflammation Value; SII: Systemic immune–inflammation index; TRV: tricuspid velocity; WBC: leukocytes.

## Data Availability

All data presented in this study are available within the article. The first author has all data used in this study.

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
