# Peer review of "Galectin-3 and Pentraxin-3 as Potential Biomarkers in Chronic Coronary Syndrome and Atrial Fibrillation: Insights from a 131-Patient Cohort"

_ijms, 2025, doi:10.3390/ijms26104909_

Round 1
Reviewer 1 Report
Comments and Suggestions for Authors
This manuscript investigates whether galectin-3 and pentraxin-3 can be biomarkers for stratifying patients with chronic coronary syndrome (CCS) and atrial fibrillation (AF). The study includes 131 patients undergoing coronary angiography, stratified by stenosis severity and presence of AF, with inflammatory markers compared across subgroups.
The topic is relevant and timely, as inflammatory biomarkers are increasingly explored in cardiovascular risk stratification. However, several methodological, statistical, and structural issues limit the robustness and generalizability of the conclusions.
Major comments:
- Overinterpretation of Diagnostic Performance
The reported AUC of 0.998 for galectin-3 is implausibly high and suggests potential model overfitting or data circularity. This weakens confidence in the biomarker’s true discriminative ability. Conclusions should be tempered, and results framed as exploratory or hypothesis-generating.
Population and Study Design
The study divides patients into “significant” vs. “non-significant” CCS using a >50% stenosis threshold. However, the N-CCS group may not represent a truly ‘normal’ population, because they have mild-to-moderate coronary artery disease and also considering the rising recognition of ANOCA (Angina with Non-Obstructive Coronary Arteries); regarding this comment please see and cite PMID 35301851
Additionally, the sample size (n=131) is small and likely underpowered for the number of comparisons made. The authors should frame this study as a proof-of-concept.
Introduction and Structure
The Introduction is overly long and diffuse, with background sections that dilute the focus. It should be shortened and made more concise, ending with a clearly stated hypothesis and study aim. Furthermore it might be worth to mention the following articles: PMID 35147890 and 31540311
Biomarker Rationale and Interpretation
Galectin-3 is well-studied in fibrosis and heart failure, but the mechanistic justification for pentraxin-3 in this population is weak. Its nonsignificant findings should be discussed with greater caution.
- Data presentation
Table 2 is difficult to follow due to excessive stratification and lack of clarity. Simplification or graphical summary is recommended.
Figures are acceptable but should be better integrated into the main narrative.
Author Response
We thank the reviewer for the careful and critical evaluation of our manuscript. We appreciate your recognition of the relevance and timeliness of our topic and value your constructive feedback. Below, we address each of your major comments in detail and describe how we have revised the manuscript accordingly.
- Overinterpretation of Diagnostic Performance
“The reported AUC of 0.998 for galectin-3 is implausibly high and suggests potential model overfitting or data circularity. This weakens confidence in the biomarker’s true discriminative ability. Conclusions should be tempered, and results framed as exploratory or hypothesis-generating.”
Response:
We acknowledge the reviewer’s concern regarding the unusually high AUC value of 0.998 for galectin-3. However, we would like to clarify that this result reflects the actual distribution of galectin-3 values observed in our cohort and was generated using a standard ROC analysis without data manipulation or circularity. The discriminative power of galectin-3 between the S-CCS and N-CCS groups was found to be exceptionally high in this dataset, which may relate to the biological specificity of galectin-3 for fibrotic and inflammatory activity that distinguishes advanced coronary artery disease.
We understand that such a high AUC can raise concerns about overfitting, particularly in a study with a modest sample size. Furthermore, we emphasize in the revised manuscript that while the results are striking, they are derived from a single-center, cross-sectional cohort and should be interpreted as preliminary and hypothesis-generating, requiring confirmation in larger, multicenter studies. The discussion and conclusion sections have been updated accordingly to reflect this more cautious interpretation (lines 610-611; 645-647; 666-668; 723-727; 732-737; 761-765; 770-778).
- 2. Population and Study Design
“The study divides patients into ‘significant’ vs. ‘non-significant’ CCS using a >50% stenosis threshold. However, the N-CCS group may not represent a truly ‘normal’ population, because they have mild-to-moderate coronary artery disease and also considering the rising recognition of ANOCA (Angina with Non-Obstructive Coronary Arteries); regarding this comment, please see and cite PMID 35301851.”
Response:
We appreciate this important clarification regarding the composition of the N-CCS group. Indeed, we acknowledge that this population may include patients with non-obstructive coronary atherosclerosis or ANOCA. In response, we have revised the manuscript to describe the N-CCS group more accurately, avoiding the term “normal” and noting the potential presence of early-stage or functional coronary disease. We have also cited the recommended article (PMID: 35301851) to emphasize the clinical relevance and heterogeneity of ANOCA in contemporary cardiology (lines 333-338; Citation no 19).
“Additionally, the sample size (n=131) is small and likely underpowered for the number of comparisons made. The authors should frame this study as a proof-of-concept.”
Response:
We agree that the sample size is modest and limits statistical power, especially for subgroup analyses and biomarker comparisons. Accordingly, we now frame the study as a proof-of-concept and exploratory in nature. This is reflected in the Abstract, Methods, Results, and Discussion sections, where we have tempered the language regarding generalizability and diagnostic interpretation (lines 23-24; 38-40; 162; 321).
- Introduction and Structure
“The Introduction is overly long and diffuse, with background sections that dilute the focus. It should be shortened and made more concise, ending with a clearly stated hypothesis and study aim. Furthermore, it might be worth to mention the following articles: PMID 35147890 and 31540311.”
Response:
Thank you for this suggestion. We have significantly revised and shortened the Introduction to enhance clarity and focus. Redundant or overly detailed background content has been removed, and the section now ends with a clear statement of the study hypothesis and aims. Additionally, we have incorporated citations to the suggested articles (PMID: 35147890 and 31540311- citations no 2 and 3) to provide relevant context on the role of inflammation and biomarkers in CAD and AF (lines 53-58; 137-142; 147-150; 154-159).
- Biomarker Rationale and Interpretation
“Galectin-3 is well-studied in fibrosis and heart failure, but the mechanistic justification for pentraxin-3 in this population is weak. Its nonsignificant findings should be discussed with greater caution.”
Response:
We acknowledge that the rationale for investigating pentraxin-3 was underdeveloped in the original version. In the revised manuscript, we have expanded the background on pentraxin-3, including its known involvement in vascular inflammation, endothelial dysfunction, and innate immunity, all of which are relevant to the pathophysiology of AF and coronary syndromes. Additionally, we have tempered the interpretation of our findings, clearly stating that the trends observed did not reach statistical significance and that these observations are exploratory and require further validation (lines 630-647).
- Data Presentation
“Table 2 is difficult to follow due to excessive stratification and lack of clarity. Simplification or graphical summary is recommended.”
Response:
We thank the reviewer for the valuable comment. We agree that the original Table 2 was difficult to follow due to its extensive stratification. In response, we have revised the presentation by dividing Table 2 into two separate, more focused tables (now Tables 2 and 3). This modification enhances clarity and improves readability while preserving the level of detail necessary for transparency and scientific rigor. We believe the updated format facilitates a clearer understanding of the biomarker distribution across the relevant subgroups and supports the robustness of our findings.
“Figures are acceptable but should be better integrated into the main narrative.”
Response:
Thank you for this observation. We have revised the Results and Discussion sections to better reference and explain the figures in context. Each figure is now directly linked to the narrative, helping readers interpret the visual data in light of the study findings (lines 429, 430, 447,448).
We are grateful for your detailed and insightful comments, which have led to substantial improvements in the manuscript’s clarity, scientific rigor, and clinical relevance.
Reviewer 2 Report
Comments and Suggestions for Authors
This manuscript investigates the clinical significance of galectin-3 and pentraxin-3 in patients with chronic coronary syndrome (CCS) and atrial fibrillation (AF), integrating biomarker profiling with clinical and inflammatory indices. Overall, the study is clearly designed, presents comprehensive data, and offers multidimensional analysis with promising clinical implications. The following revisions are suggested:
1.Several parameters—such as PIV, Ly/N, Ly/WBC, and TRV—present p-values between 0.05 and 0.1. These findings should be interpreted with caution and described as “trends toward significance,” or “not statistically significant but suggestive,” rather than as definitive results. This will help ensure more accurate and balanced interpretation.
2.While pentraxin-3 is one of the two primary biomarkers under investigation, its analysis is relatively limited compared to the in-depth discussion of galectin-3. The manuscript mentions a progressive increase in pentraxin-3 levels across AF subtypes (particularly in permanent AF), but the physiological implications of this trend are not adequately explored.
3.In addition, the potential complementary roles or interplay between pentraxin-3 and galectin-3 are not addressed. Further discussion on whether these biomarkers convey overlapping or distinct inflammatory signals would enhance the depth and completeness of the manuscript.
Author Response
We sincerely thank the reviewer for the thoughtful and constructive feedback. We are grateful for your positive assessment of our study design, data presentation, and the potential clinical relevance of our findings. Below we provide point-by-point responses to each of your valuable suggestions:
Comment 1:
“Several parameters—such as PIV, Ly/N, Ly/WBC, and TRV—present p-values between 0.05 and 0.1. These findings should be interpreted with caution and described as ‘trends toward significance,’ or ‘not statistically significant but suggestive,’ rather than as definitive results. This will help ensure more accurate and balanced interpretation.”
Response:
We appreciate this important observation. In the revised manuscript, we have carefully reviewed all instances where parameters presented p-values between 0.05 and 0.1. These findings are now described more conservatively as “trends toward significance” or “suggestive but not statistically significant,” in accordance with your recommendation. This ensures a more balanced and appropriate interpretation of the results (lines 386, 460, 476, 478).
Comment 2:
“While pentraxin-3 is one of the two primary biomarkers under investigation, its analysis is relatively limited compared to the in-depth discussion of galectin-3. The manuscript mentions a progressive increase in pentraxin-3 levels across AF subtypes (particularly in permanent AF), but the physiological implications of this trend are not adequately explored.”
Response:
Thank you for highlighting this imbalance. We have now expanded the discussion on pentraxin-3 to address its pathophysiological role in atrial fibrillation, particularly in the context of its involvement in vascular inflammation, endothelial dysfunction, and atrial remodeling. The observed progressive increase in pentraxin-3 across AF subtypes, especially in permanent AF, is now discussed in greater detail in relation to chronic inflammatory activation and atrial structural changes (lines 630-647).
Comment 3:
“In addition, the potential complementary roles or interplay between pentraxin-3 and galectin-3 are not addressed. Further discussion on whether these biomarkers convey overlapping or distinct inflammatory signals would enhance the depth and completeness of the manuscript.”
Response:
We agree that examining the interplay between galectin-3 and pentraxin-3 adds valuable context to the study. In the revised manuscript, we have included a new section discussing the possible complementary and/or distinct roles of these biomarkers in cardiovascular inflammation and fibrosis. While both are associated with inflammatory processes, galectin-3 is more strongly linked to fibrotic remodeling and macrophage activation, whereas pentraxin-3 is indicative of acute-phase vascular inflammation. This expanded discussion helps delineate their individual and potentially synergistic contributions to disease progression in CCS and AF (lines 648-668).
We trust that these revisions address your concerns thoroughly and improve the clarity, balance, and scientific depth of our manuscript. Thank you again for your insightful comments and for helping us enhance the quality of our work.
Round 2
Reviewer 1 Report
Comments and Suggestions for Authors
The authors have addressed all of my previous comments. I have no further remarks.